# Effect of the Glass Fiber Content of a Polybutylene Terephthalate Reinforced Composite Structure on Physical and Mechanical Characteristics

**DOI:** 10.3390/polym14010017

**Published:** 2021-12-22

**Authors:** Oumayma Hamlaoui, Olga Klinkova, Riadh Elleuch, Imad Tawfiq

**Affiliations:** 1Laboratoire QUARTZ EA7393, ISAE-Supméca Institut Supérieur de Mécanique de Paris, 93400 Saint-Ouen, France; hamlauioumeyma@gmail.com (O.H.); imad.tawfiq@isae-supmeca.fr (I.T.); 2Laboratoire des Systèmes ElectroMécaniques (LASEM), Ecole Nationale D′Ingénieurs de Sfax, Sfax 3038, Tunisia; riadh.elleuch@gnet.tn

**Keywords:** composite, polybutylene terephthalate (PBT), glass fiber content, characterization, properties, damage

## Abstract

This work presents the influences of glass fiber content on the mechanical and physical characteristics of polybutylene terephthalate (PBT) reinforced with glass fibers (GF). For the mechanical characterization of the composites depending on the GF reinforcement rate, tensile tests are carried out. The results show that increasing the GF content in the polymer matrix leads to an increase in the stiffness of the composite but also to an increase in its brittleness. Scanning Electron Microscope analysis is performed, highlighting the multi-scale dependency on types of damage and macroscopic behavior of the composites. Furthermore, flammability tests were performed. They permit certifying the flame retardancy capacity of the electrical composite part. Additionally, fluidity tests are carried out to identify the flow behavior of the melted composite during the polymer injection process. Finally, the cracking resistance is assessed by riveting tests performed on the considered electrical parts produced from composites with different GF reinforcement. The riveting test stems directly from the manufacturing process. Therefore, its results accurately reflect the fragility of the material used.

## 1. Introduction

Reinforced composites are nowadays widely used in multifunctional industrial products, such as electrical and electronic components. The choice of composite material is closely related to its use, more precisely the choice of the matrix, the fibers and the reinforcement rate is based on their mechanical and physical characteristics. Moreover, the choice of plastic composite material for the injection molding process is a key step in new product development and integration. This choice is crucial and will lead to the optimum injection process parameters, assembly methods, and the determination of product use.

This study focuses on the assessment of the impact of the glass fiber (GF) content of PBT/GF composite on the mechanical and physical characteristics of electrical component parts. The riveting operation of the metal pin in the composite body leads to an overload of the material. This most often causes a crack in the neck of the composite part as illustrated in Figure 1.

The reinforcement, its nature, its form (powder, short fibers, etc.) and the type of stress during assembly are inter-correlated to the failure mode of a composite material. The reinforcement rate impacts the behavior of the material [1,2,3,4]. The main purpose of short GF reinforcement is to enhance the mechanical characteristics in terms of rigidity and mechanical resistance. Many studies have been conducted to evaluate the impact of GF on the mechanical and physical characteristics of composites. GF reinforcement enhances the mechanical characteristics of reinforced polymers in terms of Young’s Modulus, tensile strength and elastic limit, confirmed for PA66 (Polyamide 66), PES (Polyethersulfone), PLA (Polylactic Acid), and PVC (PolyVinyl Chloride) [3,5,6,7]. Although GF improves mechanical characteristics it reduces elongation at break which impacts the ductility of the material, making it incapable of being subjected to greater deformation although capable of tolerating larger stress ranges [2,5,8,9].

Furthermore, rupture mechanisms are related to GF dispersion and orientation in the structure whether it is horizontal or vertical to the injection direction. This can impact GF rupture in the structure or more often their pull out from the matrix [7].

Not only mechanical properties are influenced by the presence of GF; the reinforcement is also used for safety reasons. The type of the reinforcement impacts the flammability of the composites. GF prevents and retards composite flammability. Flame retardancy is related to chemical composition, the reinforcement rate and shape. Composites are characterized according to international references using one of the two recommended methods: the glow wire test or UL94 [10,11,12,13,14,15,16,17,18]. The GF reinforcement rate for composites has an impact on fluidity. Increasing the reinforcement rate decreases the fluidity of the GF/Polymer composites which has an impact on the surface of the injected molding [19,20]. To determine the impact of the GF reinforcement rate on the fluidity and surface aspect, the melt flow index (MFI) of the composite is used.

In this paper, the impact of the GF rate on the mechanical properties of the composite, such as Young’s modulus, tensile strength, elastic limit, and elongation at break is investigated. Additionally, to study the impact of the reinforcement rate on the flammability of the GF/PBT composite, the glow wire test was carried out according to standards [10,11,12] on different plates injected with the GF/PBT composite with different reinforcement rates. Flame height was measured for the 30 s test and the inflamed aspect was explored. The riveting test was carried out on high reinforced GF/PBT composites at 30%, 25%, and 20% to link the GF rate to the overall failure percentage. Flammability and fluidity were assessed to optimize the GF rate and keep the same flammability properties for safety reasons and the same fluidity specifications to ensure a stable injection molding process. The SEM observation was carried out on pure PBT Resin, 5% and 30% reinforced PBT to study the ruptures of the GF/PBT composite.

The main aim of this work is to study the influence of the GF reinforcement rate on the most requested properties of the composite based on PBT matrix and used for the production of electrical parts.

## 2. Materials and Methods

### 2.1. Materials

PBT (Polybutylene terephthalate) polymer reinforced with different GF (glass fibers) contents, obtained by mixing commercialized VALOX 420 (Sabic Ltd., Chongqing, China) (30 wt% GF/PBT) and commercialized VALOX 310 (Sabic Ltd., Chongqing, China) (unreinforced PBT), presented in granular form. The used short glass fibers have the following mechanical and morphological properties:Young modulus E (MPa): 78,500;Tensile strength R_m_ (MPa): 1950;Density (g/cm^3^): 2.55;Average diameter (μm): 13.8.

Table 1 summarizes the different proportions of mixing percentages. The mixing process is performed by a mixing machine from Meter Mix Systems Ltd. (Rushden, UK) for 30 min to ensure the homogeneity of GF dispersion in the material during the injection process.

### 2.2. Methods

#### 2.2.1. Mechanical Characterization

##### Specimen Preparation

The material configurations studied are dried at 120 °C for 4 h with the dehumidifier to ensure a maximum moisture rate lower than 0.02%. The specimens are then injected by a Kraussmafei 50-ton injection machine at 800 bar and 275 °C using a dosage volume of 15 cm^3^ dosed for 2.8 s, injected for 0.8 s and subjected to a locking force of 300 bar and a counter pressure of 100 bar. The diagram of the filling process of the specimen for the tensile test is highlighted in Figure 2. The injected material is subjected to a cooling cycle of 10 s at 60 °C at a pressure maintained at 400 bar. The specific tool is used to produce the rectangular plates with the dimensions 125 mm × 13 mm × 3 mm. The tensile samples are then obtained from the rectangular plates, using a CIF Techno-drill 3 XL milling machine with a diamond bur with a 2.5 mm diameter. The advancing speed was set to 5 mm/s and the rotation frequency was set to 22,000 rpm with a depth of cut equal to 1 mm. Three cuts were performed for the 3 mm thick specimen.

##### Tensile Test

Mechanical characterization static tensile testing was conducted using a Zwick/Roell Z100 tensile machine (Metz, France) with a 100 kN force cell and optical 3D Motion technique for strain capture. The tests were carried out in accordance with the ISO 527-2 [21] standard. The standardized 1B type specimens were fixed between the two jaws of the tensile machine and two white stickers were glued on the useful part of the specimen, as shown in Figure 3. The tensile tests were run at room temperature with a test rate of 5 mm/min. LED lamps were used to light the surface to ensure the camera detects the displacement shadow.

##### Microscopic Analysis

The Scanning Electron Microscope (SEM) observations were made using a Jeol JSM-6010 PLUS/LA (Croissy-sur-Seine, France). SEM specimens were cut transversally from the tensile test specimens, coated with resin and then polished. The observations were made on different specimens with different rates: 0% (unreinforced PBT), 5% and 30% GF reinforced PBT, and observed under low pressure. The observations were made at different scales to better capture the damage mechanics.

#### 2.2.2. Flammability Test

Glow wire tests were performed based on IEC 60695-2-13 [10], IEC 60335-1 [11] and IEC 60695-2-11 [12] standards and according to the application of the composite part (fuse holder). The glow wire test set-up, which is based on the direct contact of a glowing wire at 850 °C assessed by a K-type thermocouple for 30 s and measuring the flame height during the test. The test is considered valid when no flowing material is detected either during or at the end of the test, and the flame extinguishes itself after the glow wire is removed.

#### 2.2.3. Fluidity Test

The fluidity test was performed according to the ISO1133-1 [22] standard by choosing the material flow index measurement by mass MFR (material flow rate). In total, 100 g of material was used. The material fusion temperature was set at 250 °C.

#### 2.2.4. Riveting Test on the Considered Electrical Part

Figure 4 shows the pin riveting operation of the composite part and the riveting tool used. The composite part is fixed to the riveting tool (Figure 4a). Then, a metallic pin is placed in the upper part of the fuse (Figure 4b) and finally, the riveting operation is performed by the vertical movement of the riveting tool (Figure 4c), with single, double, and triple strokes depending on the GF content of the composite.

## 3. Results

### 3.1. Tensile Test Results

Tensile tests of the original material (30 wt% GF/PBT) were carried out at 5 and 20 mm/min. Figure 5 shows the conventional stress–strain diagrams with no influence of the testing velocity. Therefore, the testing velocity of 5 mm/min was chosen for further analysis.

The tensile test results were compared with the values from the technical sheets for both the PBT polymer (0% GF) and the VALOX 420 (30 wt % GF/PBT). The results of the tensile tests showed a change in the material behavior depending on the proportion of the short glass fibers in the composite (Figure 6) from ductile (pure polymer, 0% GF) to very brittle (30 wt% GF). This was reflected in the significant decrease in the elongation at break from 0.15 for the specimen with 0% reinforcement to 0.015 for the specimen with 30% reinforcement.

The mean curves plotted in Figure 6 were obtained from 5 tests for each configuration. The mechanical characteristics (Young modulus E, elastic limit R_e_, elongation at break A and tensile strength R_m_) with the respective standard deviation are summarized in Figure 7.

In Table 2 the measured values of the different mechanical properties (Young modulus E, Yield strength R_e_, elongation at break A and tensile strength R_m_) of the studied composites depending on the GF reinforcement rate are presented. Figure 6 show the evolution of these parameters. The results point that the PBT/GF composite mechanical properties are significantly impacted by increasing the GF reinforcement rate in the structure. The Young’s modulus increased to 77.33% when increasing the GF reinforcement rate to 30%. The elastic limit and the tensile strength increased by 65.12% and 63.28%, respectively. Elongation at break was inversely impacted by increasing the GF reinforcement in the composite structure and dropped 14 times when reaching a reinforcement rate of 30%. The conventional curves in Figure 6 and the mechanical characteristics measured in Figure 7, all confirm the increase in brittleness when increasing the GF reinforcement rate in the composite structure.

### 3.2. Microscopic Analysis

Macroscopic behavior, damage kinetics, and failure mode strongly depend on the GF content of the composite. SEM analysis was performed on all the configurations but only two boundary configurations were chosen (30% GF/PBT and 5% GF/PBT) to demonstrate GF spatial distribution and typical damage that occurred in comparison to the PBT polymer. Figure 8 illustrates SEM micrographs of cross-sections of tensile test samples of 30% GF/PBT, the more brittle composite structure. Injection was performed in the middle of the plate, in out of plane direction. The fiber space dispersion can be seen on the left-hand side in Figure 8a. The zone of fiber bounded by the orange rectangle presents the injection path in the center of the image, which leads to fiber orientations depending on the injection parameters.

As shown in Figure 8 we noticed that the increase in micro-cracks induces the rupture of the tensile specimen. This phenomenon was generated by three mechanisms:The cracks of the GF itself;The release of GF from the PBT matrix; andThe failure of the matrix.

Figure 9 and Figure 10 show the SEM micrographs of a cross-section of a tensile sample of 5% GF/PBT. The fiber distribution is much more homogeneous in the matrix bulk. The injection effect is barely visible for this GF rate. The SEM observation for the ruptures for a tensile specimen injected with a 5% GF/PBT composite presented in Figure 9 and Figure 10 shows more matrix cracks and less GF cracking. The energy required to rupture the GF is higher than that required to rupture the matrix [7]. Thus, the tensile stress increases as the GF reinforcement rate increases. However, elongation at break is lower since there is more GF reinforcement which reduces the ductility and increases the fragility of the structure. For the low GF reinforced material, we have less GF and more dispersion which causes rupture under low stress with considerable elongation at break.

Figure 11 shows SEM micrographs of a cross-section of a tensile sample of pure resin PBT.

### 3.3. Flammability Test Results

Assessing the flammability of each material with different GF reinforcement rates is crucial, especially because GF reinforcement is used as a flame-retardant mechanism and the composite part is a fuse holder that can be ignited if there is a short circuit. Based on standards IEC 60695-2-13 [10], and IEC 60695-2-11 [12] the glow wire test is considered valid if there is no material flowing for 30 s while the glow wire is in contact, and self-extinction occurs after the removal of contact with the glow wire. An additional safety parameter is added: flame height must not exceed 8 cm at maximum, in order not to exceed the total length of the part, prevent its total ignition and burning. Figure 12 summarizes the flammability test results repeated 5 times for each configuration. The flame height is higher for the composite with a lower GF reinforcement rate. Flame height reaches 36.4 mm for the pure resin; however, from GF reinforcement rate of 20%, the flame height decreases to reach 9.2 mm. The flames heights became lower and lower; 5.4 mm for the 25% GF/PBT composite and 4.4 mm for the 30% GF/PBT composite. No flowing material and flame self-extinction were observed for any of the conditions after contact with the glow wire was removed.

Figure 13 highlights the contact images versus the GF reinforcement rate 30 s after the test. No material flow is visible on the structure, but the damage kinetics depends on the GF content of the materials tested. The pure PBT matrix (no reinforced structure) had a debouching penetration, with a visible white zone in the middle of the image. The damaged ignited zone is wider for higher GF reinforced composites. From 10% GF/PTB upwards, debouching penetration no longer occurs, thus the GF reinforcement impacts the mechanical characteristics of the structure and acts as a flame retardant.

### 3.4. Fluidity Test Results

Figure 14 highlights the increase in the GF rate in a GF/PBT composite from 15% to 30%; the MFI decreased from 12.2 to 10.1 g/10 min. These results reflect the impact of GF on the moldability of the material. The decrease in MFI impacts the surface quality of the part and causes the surface defects, burns, and micro burns shown in the Figure 15.

### 3.5. Riveting Test Results

Riveting tests were performed to study the impact of GF on the cracking of components. The GF rate was adjusted from 20% to 25% and 30% during the injection of the part. The riveting operation was performed on specimens with different conditions related to the number of strokes, as shown in Table 3. The riveting operation is performed with an imposed displacement through a stroke. The displacement of 7 mm is settled during the production launch. The load is 15 N/m^2^. For 20% GF/PBT no failure percentage was found for any of the test conditions. For the composite reinforced with 25% GF the failure percentages were 7%, 10%, and 20%, respectively. For 30% GF the failure percentages were 8%, 13%, and 30% for the tests done with simple, double, and triple strokes. The number of stroke and the GF reinforcement rate increase the failure percentage of riveted specimens.

## 4. Discussion

The impact of velocity was observed differently on the GF/PA (polyamide) composite where the variation of velocity had an impact on the strain rate, thus the Young’s modulus, elastic limit, tensile strength and elongation at break [5,8]. The impact of varying the displacement rate also shows the visco-plastic behavior of a GF/PP (polypropylene) composite [1]. After increasing the GF rate in the structure, same behavior as PBT/GF composite was noted for PA66, polyester and rubber-toughened nylon 6, where there was an increase in fragility expressed by the increase in the tensile strength and Young modulus versus a decrease in the elongation at break [2,5,8,9]. The same trend of ameliorating the mechanical properties and especially increasing tensile strength when increasing the percentage of GF reinforcement was found in the literature [6,7,9,23,24]. PLA (Poly(Lactic)Acid) reinforced by different GF rates showed a linear decrease in elongation at break and a linear increase in Young’s modulus and tensile strength [7]. The Young’s modulus of Polyethersulfone/GF, studied by [6], increased from 19 GPa for 50/50 GF/Polymer reinforcement rate to 23 GPa for 70/30 GF/Polymer.

SEM micrographs of the transverse surface of a fractured tensile test sample shows that the defects are generated in the interface (pull out, fiber breakage) that generates the fracture process. Similar failure mechanisms were reported for composite based of PLA, Polyethersulfone, PA66, and PVC polymers reinforced by GF [3,5,6,7]. According to the literature the shape and aspect of the ignited area depends on the type and rate of reinforcement [4].

The same results have been confirmed for PP whose surface properties degraded when increasing the fill percentage in GF and talc [20]. The GF reinforcement rate impacts fluidity during the injection process, thus the surface roughness, part aspect, and dimensions are dependent on this material property [7,20]. Thus, when increasing the GF rate in a GF/PLA [5,19] and GF/PP [20] the material fluidity is inversely affected. GF/PC studied by [19] reported that increasing the GF rate from 10% to 20% decreased the MFI (Melt Flow Index) from 10 to 6 g/10 min.

## 5. Conclusions

GF reinforcements enhance the mechanical properties of the composite based on PBT matrix, by increasing its Young’s modulus, yield strength and elastic limit. However, high GF reinforcement rates in the structure causes a decrease in the elongation at break, which means an increase in the brittleness of the composite. The mechanical behavior of the studied composite depending on the GF reinforcement rate was confirmed by SEM observations of surface ruptures. So pure resin PBT showed only matrix cracks that confirmed the ductility of the material. However, reinforced composites showed matrix cracks, GF pull out, and cracks.

Furthermore, the GF reinforcements influence the flammability of the composite. The results improve that the flame height decreased when the GF reinforcement rate increased.

Finally, the presented work shows that the GF reinforcement rate also affects the fluidity of the composite in the injection molding process. The results of the fluidity test showed that MFI decreased when increasing the GF rate in the composite which led to the appearance of micro-faults on the surface of the injected GF/PBT composite component.

## Figures and Tables

**Figure 1 polymers-14-00017-f001:**
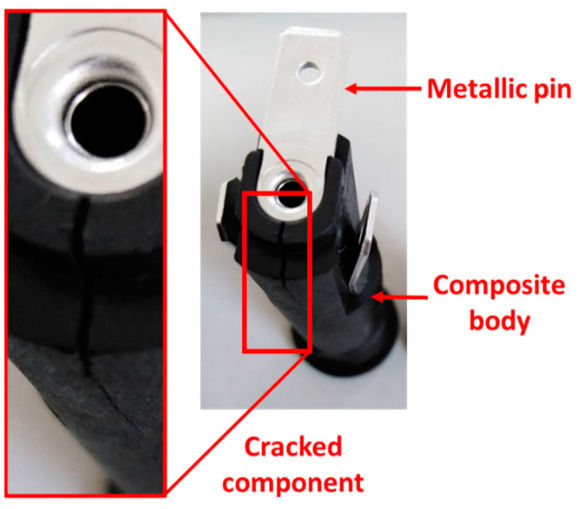
Crack in the considered electrical component.

**Figure 2 polymers-14-00017-f002:**
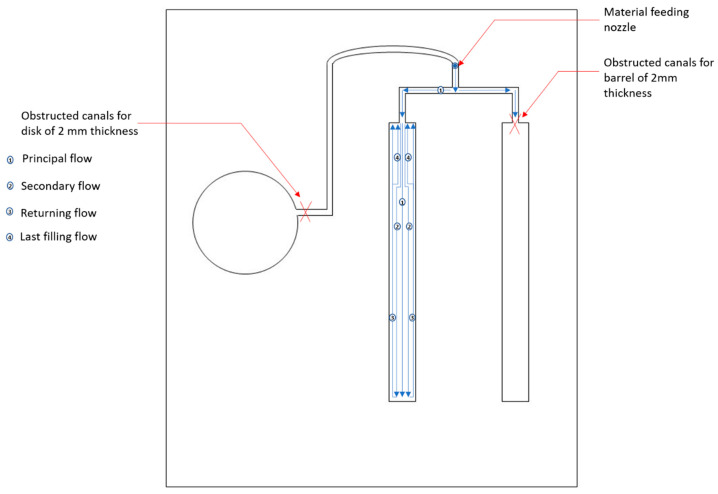
Diagram of the filling process of the specimen for the tensile test.

**Figure 3 polymers-14-00017-f003:**
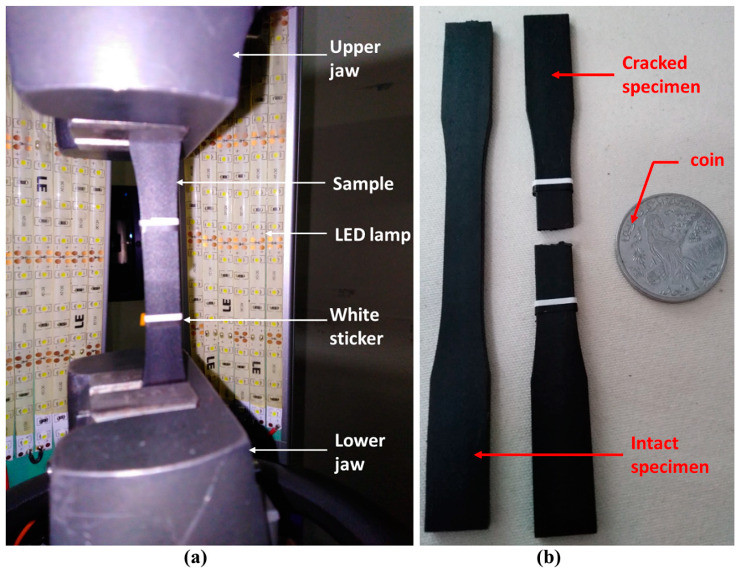
(**a**) Tensile test set-up; (**b**) Sample before and after test.

**Figure 4 polymers-14-00017-f004:**
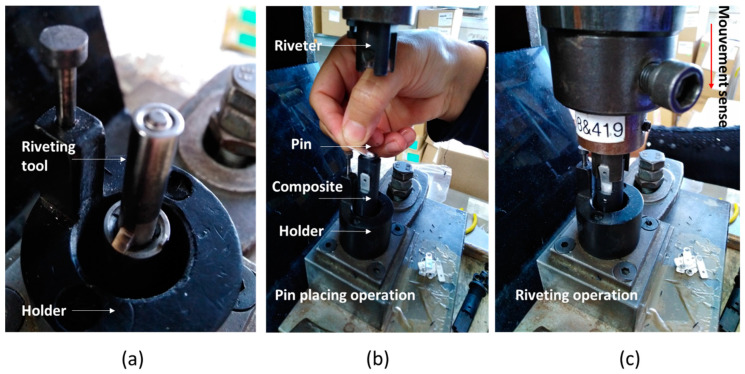
(**a**) Riveting bench; (**b**) Pin placing operation; (**c**) Pin riveting operation of composite.

**Figure 5 polymers-14-00017-f005:**
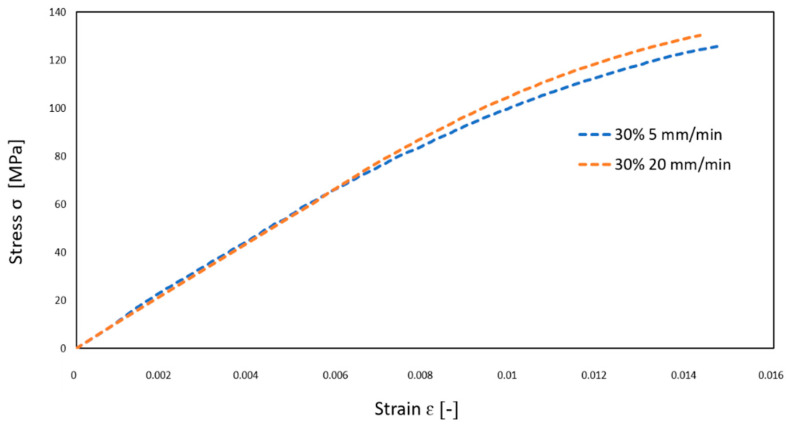
Conventional stress–strain diagrams of 30% GF/PBT tested at 5 and 20 mm/min.

**Figure 6 polymers-14-00017-f006:**
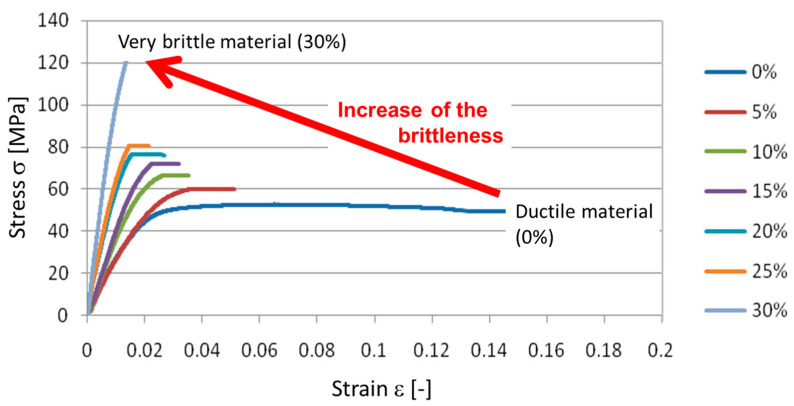
Conventional mean stress–strain diagrams (5 tests/conditions) of the composite materials reinforced with different GF reinforcement rates.

**Figure 7 polymers-14-00017-f007:**
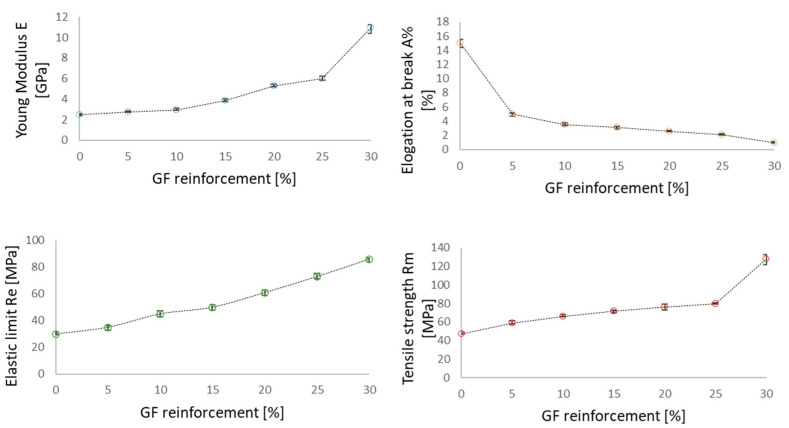
Mechanical characteristics of the composite materials vs. GF reinforcement rate with standard deviation bars.

**Figure 8 polymers-14-00017-f008:**
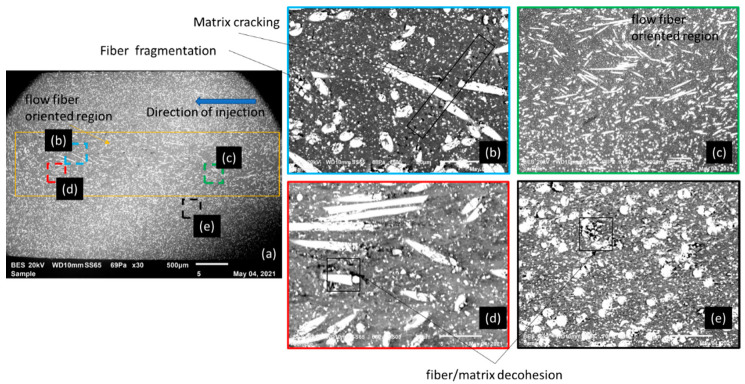
SEM micrographs of the transverse surface of a fractured tensile sample with 30% GF/PBT composite, 20 kV, 10 mm working distance, 69 Pa: (**a**) BES macro-image with flow fiber-oriented region; (**b**–**d**) zoom of different regions in the zone (**a**); (**e**) homogeneous zone.

**Figure 9 polymers-14-00017-f009:**
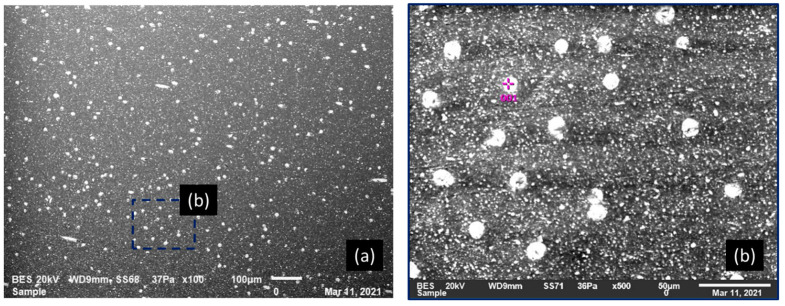
SEM micrographs of the transverse surface of a fractured tensile test sample with 5% GF/PBT composite, 20 kV, 9 mm working distance, 36 Pa: (**a**) BES macro-image; (**b**) fiber distribution on PBT matrix.

**Figure 10 polymers-14-00017-f010:**
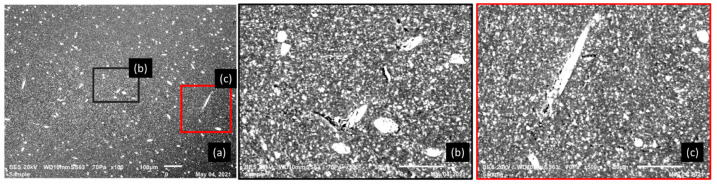
SEM micrographs of the transverse surface of a fractured tensile test sample with 5% GF/PBT composite, 20 kV, 10 mm working distance, 70 Pa: (**a**) BES macro-image; (**b**,**c**) zoom of different regions in the zone (**a**).

**Figure 11 polymers-14-00017-f011:**
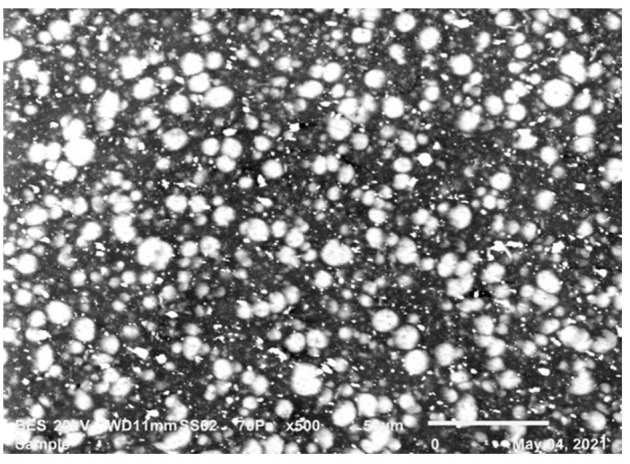
SEM micrograph of PBT specimen, 20 kV, 10 mm working distance, 70 Pa.

**Figure 12 polymers-14-00017-f012:**
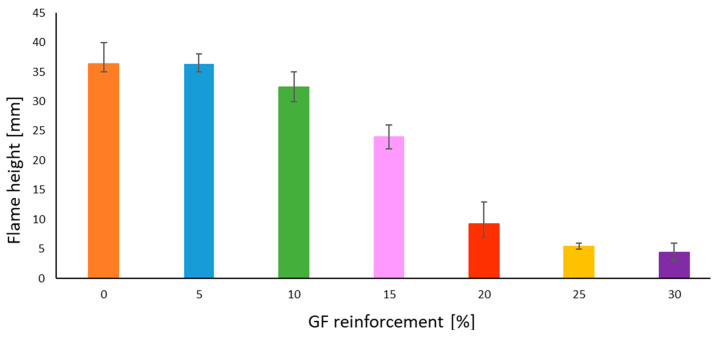
Bar chart of flame height vs. GF reinforcement rate with standard deviation bars.

**Figure 13 polymers-14-00017-f013:**
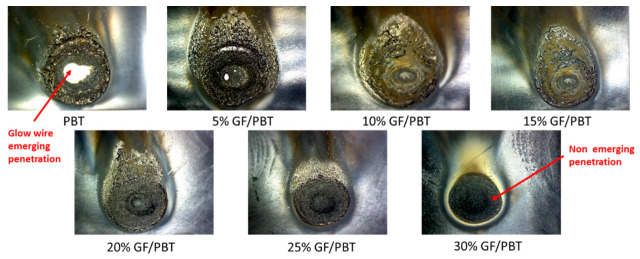
Appearance of the ignited surface vs. GF reinforcement rate.

**Figure 14 polymers-14-00017-f014:**
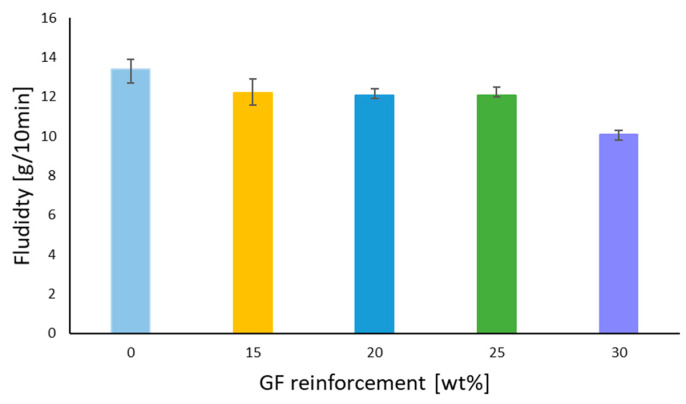
Fluidity vs. GF reinforcement rate.

**Figure 15 polymers-14-00017-f015:**
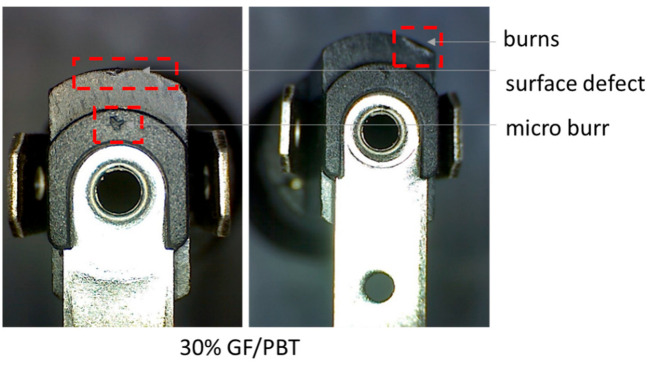
Optical image of micro-faults on 30% GF/PBT composite.

**Table 1 polymers-14-00017-t001:** Mixing percentage for composite materials with different GF contents.

Composites under Study (Required GF Reinforcement wt%)	0	5	10	15	20	25	30
Quantity of PBT Resin, g	100	500	200	100	50	20	0
Quantity of PBT/GF reinforced of 30%, g	0	100	100	100	100	100	100

**Table 2 polymers-14-00017-t002:** Impact of GF reinforcement rate on the variation of the mechanical properties.

Material	Young Modulus E (MPa)	Yield Strength Re (MPa)	Elongation at Break A (%)	Tensile Strength Rm (MPa)
PBT	-	-	-	-
5 wt% GF/PBT	+10.28%	+14.29%	−200.00%	+20.34%
10 wt% GF/PBT	+16.45%	+33.33%	−328.57%	+28.79%
15 wt% GF/PBT	+36.05%	+40.00%	−383.87%	+34.72%
20 wt% GF/PBT	+53.20%	+50.82%	−476.92%	+38.16%
25 wt% GF/PBT	+58.82%	+58.90%	−614.29%	+41.25%
30 wt% GF/PBT	+77.33%	+65.12%	−1400.00%	+63.28%

**Table 3 polymers-14-00017-t003:** Riveting test results as a function of GF reinforcement rate.

GF wt%	Simple Stroke		Double Stroke		Triple Stroke	
	Number of Specimens	Failure Percentage (%)	Number of Specimens	Failure Percentage (%)	Number of Specimens	Failure Percentage (%)
20	25	0	15	0	10	0
25		4		7		10
30		8		13		30

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
