# Peer review of "Effect of the Glass Fiber Content of a Polybutylene Terephthalate Reinforced Composite Structure on Physical and Mechanical Characteristics"

_polymers, 2021, doi:10.3390/polym14010017_

Round 1
Reviewer 1 Report
The work described in the paper is interesting and well written, authors have defined the experimental aspects very well but there are some deficiencies before acceptance, few points are –
- Title and abstract is OK but please expand PBT in title and keywords as well. Moreover, please prepare a section of abbreviations and summarize all the abbreviations used in the paper. What is the target application of the work?
- In materials section, please provide morphological details of the glass fiber, its physical and mechanical properties?
- In introduction, please refer 3-4 papers from Polymers-MDPI in form of table and highlight the advancement of this work over previous publications. How this work is important and what is its individual contribution in this field of research?
- In Figure 12, x-axis, is the content of GF wt% or vol%. it is confusing? Moreover, the addition of GF in polymer matrix makes the composites brittle and stiff? Please comment on this aspect?
- All the Figures have poor quality and poor resolution. Please replot all images in high resolution.
- The conclusion of the paper need to be improved. The outcome of the paper need to be highlighted? Why this work is important? what are its future outlook? Good Luck !
Author Response
- Title and abstract is OK but please expand PBT in title and keywords as well.
This is done (see revised manuscript).
- Moreover, please prepare a section of abbreviations and summarize all the abbreviations used in the paper. What is the target application of the work?
The list of abbreviations is added in the paper after the conclusion (see revised manuscript).
The main target application of the work, is as mentioned in the introduction, the properties of the composite used for the production of the electrical component presented in Figure 1.
- In materials section, please provide morphological details of the glass fiber, its physical and mechanical properties?
Mechanical and morphological properties of the used short glass fibres:
- Young modulus E (MPa): 78500
- Tensile strength Rm (MPa): 1950
- Density (g/cm3): 2.55
- Average diameter (mm): 13.8
These properties are mentioned in the text (Chapter 2.1. Materials).
- In introduction, please refer 3-4 papers from Polymers-MDPI in form of table and highlight the advancement of this work over previous publications. How this work is important and what is its individual contribution in this field of research?
We have added the following two papers form Polymers-MDPI in the discussion to indicate the changing of the mechanical properties of composites by the addition of short glass fibres, which presents the main outcome of the paper:
- Abousnina R., Ibrahim Alsalmi H., Manalo A., Lim Allister R., Alajarmeh O., Ferdous W., Jlassi K. "Effect of Short Fibres in the Mechanical Properties of Geopolymer Mortar Containing Oil-Contaminated Sand", Polymers 2021, 13(17), 3008; https://doi.org/10.3390/polym13173008 - 05 Sep 2021
- Stadler G., Primetzhofer A., Jerabek M., Pinter G., Grün F. "Investigation of the Influence of Viscoelastic Behaviour on the Lifetime of Short Fibre Reinforced Polymers", Polymers 2020, 12(12), 2874; https://doi.org/10.3390/polym12122874 - 30 Nov 2020
- In Figure 12, x-axis, is the content of GF wt% or vol%. it is confusing? Moreover, the addition of GF in polymer matrix makes the composites brittle and stiff? Please comment on this aspect?
The content of GF is in wt%. This information is added in Table 1, Table 3 as well as in Table 4 and is valid for the whole work.
GF in polymer matrix makes the composites both brittle and stiff. But the effect of the increase of the brittleness is more important for this work.
- All the Figures have poor quality and poor resolution. Please replot all images in high resolution.
We apologize for the poor quality of the pictures in the first version. This is because we made the file as small as possible. In the revised manuscript, we have replotted all figures in higher resolution.
- The conclusion of the paper need to be improved. The outcome of the paper need to be highlighted? Why this work is important? what are its future outlook? Good Luck !
We have improved and in part rewritten the conclusion.
Reviewer 2 Report
Please mention you methodology on specimen to study before explaining the results.
line 12: GF content of tested composites.. How much ?
The abstract is poorly written and this makes difficult to understand the research strategy and the discussion of the results obtained, which perhaps could be interesting. The objectives are not clearly discussed and there are some attempts to rationalize the results that appear rather speculative. Neither the abstract nor the introduction explain what the aim of the work is.
Recommend an improvement of this section by highlighting the novelty/originality of this study by using more information about the purpose of this study.
Introduction need major revision.
In introduction: Why you need to mention line 38-41 ? also figure 1. ?
Table 2: What is A% Rm and E?
Please explain Figure 12 and Table 3.
Conclude the conclusion in one paragraph
Author Response
- Please mention your methodology on specimen to study before explaining the results.
In order to clarify the methodology in the presented paper, we have improved the structure of the 2nd chapter “Materials and Methods”. The main aim of this work is to study the influence of the GF reinforcement rate on the most requested properties of the composite based on PBT matrix and used for the production of electrical parts. For that, we began with the mechanical characterisation. We described the methodology for preparing the specimens. After that, we have specified the tensile test and the microscopic observation of the fracture surfaces to study the damage mechanics. Furthermore, flammability tests as well as fluidity tests are carried out. Finally, riveting tests are done on the considered electrical parts produced from composites with different GF reinforcement.
- line 12: GF content of tested composites. How much?
In this work, we have varied the GF reinforcement rate between 0 wt% (only PBT matrix without reinforcement) and 30 wt% in 5 wt% steps (see Table 1).
- The abstract is poorly written and this makes difficult to understand the research strategy and the discussion of the results obtained, which perhaps could be interesting. The objectives are not clearly discussed and there are some attempts to rationalize the results that appear rather speculative. Neither the abstract nor the introduction explain what the aim of the work is.
We have improved both the abstract and the introduction. In the introduction, we have defined clearly the main objective of this work.
- Recommend an improvement of this section by highlighting the novelty/originality of this study by using more information about the purpose of this study.
This is done.
- Introduction need major revision.
We have improved the introduction.
- In introduction: Why you need to mention line 38-41? also figure 1.?
These sentences and the picture present the main target application of the work. The scientific problem of this work is derived from the failure of the electrical part presented in Figure 1.
- Table 2: What is A% Rm and E?
E (MPa): Young modulus
A (%): Elongation at break
Rm (MPa): Tensile strength
These designations are added in the tables 2 and 3 and highlighted with a yellow marking (see revised manuscript)
- Please explain Figure 12 and Table 3.
“In Table 3 the measured values of the different mechanical properties (Young modulus E, elastic limit Re, elongation at break A and tensile strength Rm) of the studied compo-sites depending on the GF reinforcement rate are presented. Figure 12 show the evolution of these parameters. The results point that the PBT/GF composite mechanical properties are significantly impacted by increasing the GF reinforcement rate in the structure.”
This is included into the text and highlighted with a yellow marking.
- Conclude the conclusion in one paragraph
We have improved and in part rewritten the conclusion.